# Skin layer-specific spatiotemporal assessment of micrometabolism during wound angiogenesis
Weiye Li [1,2,4], Yu-Hang Liu [1,2,4], Fumimasa Kubo [3], Sabine Werner [3] & Daniel Razansky [1,2] ✉

Proper oxygen delivery through the microvasculature to injury site is essential to ensure the metabolic cascade during wound healing. Adaptation of vascular structure and oxygenation is key to unravel the regulation of blood perfusion, oxygen distribution and new tissue formation. Yet, visualizing micrometabolic responses at large scale in unperturbed living tissue remains challenging. We studied full-thickness excisional wounds in the mouse dorsal skin in vivo using ultrasound-aided spectroscopic large-scale optoacoustic microscopy. Skin layer-specific vascularization is visualized at capillary resolution over centimeter-scale field-of-view in a non-invasive, label-free manner. Different vascular parameters, including oxygenation, diameter and its irregularity, tortuosity and angular alignment, show distinct spatial and temporal variations. Elevated oxygenation is manifested close to the wound at day 4 with the trend accompanied by reduction in diameter over time. Angular alignment increases over time, indicating a more directed blood supply towards the wound. Our observations indicate that wound angiogenesis initiates as capillary sprouting with enlarged newborn vessels and elevated oxygenation around the wound, with the vessels normalizing in size and oxygenation during remodeling. Our study provides insight into micrometabolic profiles surrounding the healing wound, setting the stage for preclinical studies on oxygen delivery mechanisms in pathological skin conditions and during pharmacological interventions.

Oxygen plays a critical role in tissue healing processes[1–4]. Proper oxygen supply through the microvascular network prevents wounds from infection, sustains cell metabolism and promotes wound contraction. Disrupted oxygen delivery impairs the precisely programmed wound healing processes, leading to complications such as chronic, non-healing ulcers that cause an enormous healthcare and economic burden[5–7]. A comprehensive understanding of oxygen delivery mechanisms thus serves as a basis for devising effective treatment strategies, whilst non-invasive in vivo microscopic imaging provides important insights to advance our knowledge on microvascular oxygenation dynamics.

Despite the availability of a broad spectrum of intravital microscopy techniques, sensing oxygen delivery within the intricate microvascular network remains challenging. Confocal microscopy has been used to investigate angiogenesis dynamics during wound healing[8], chiefly attaining structural analysis within a sub-millimeter field-of-view (FOV). Likewise, two-photon phosphorescence lifetime microscopy has been demonstrated

to measure partial pressure of oxygen in the bone marrow[9] and cerebral vasculature[10,11] in live animals, but the technique requires repeated administration of exogenous oxygen probes in longitudinal imaging scenarios and similarly suffers from small FOV. Optical coherence microscopy (OCM) has also been used to study microvascular dynamics during wound healing[12], and the recent development of visible-light OCM further enables oxygen saturation (sO$_2$) measurements in retinal vasculature[13]. However, the challenging combination of high spatial resolution and large FOV hinders OCM from resolving capillary sO$_2$ over centimeter-scale areas essential for comprehensive evaluation of the wound angiogenesis processes.

Optoacoustic (OA) microscopy is uniquely equipped with longitudinal microvascular imaging capacity owing to its excellent intrinsic (label-free) functional contrast stemming from optical absorption by hemoglobin[14]. The technique has been shown suitable for visualizing and quantifying microvascular structures during wound healing[15–17]. A dual-wavelength imaging

[1]Institute for Biomedical Engineering and Institute of Pharmacology and Toxicology, Faculty of Medicine, University of Zurich, 8057 Zurich, Switzerland. [2]Institute for Biomedical Engineering, Department of Information Technology and Electrical Engineering, ETH Zurich, 8093 Zurich, Switzerland. [3]Institute of Molecular Health Sciences, Department of Biology, ETH Zurich, 8093 Zurich, Switzerland. [4]These authors contributed equally: Weiye Li, Yu-Hang Liu. ✉e-mail: daniel.razansky@uzh.ch

approach has further been employed to track microvascular oxygenation dynamics during wound healing[18], yet previous studies were limited to semi-2D assessment of superficial lesions in the mouse ear over a small (0.8 mm diameter wound) FOV, which is inadequate for studying more biologically relevant targets such as full-thickness wounds in the dorsal skin.

Here, we present a longitudinal study on the dynamics of microvascular $sO_2$ and the associated structural adaptations during cutaneous wound healing. Full-thickness 5 mm diameter wounds were generated on the dorsal skin of mice and monitored over the course of 10 days post injury in a non-invasive and label-free manner. Volumetric skin layer segmentation is achieved with an ultrasound-aided large-scale OA microscopy (uLSOM) system, enabling layer-specific analysis of wound angiogenesis. Spatiotemporal changes of microvascular $sO_2$, diameter and its irregularity, tortuosity and angular alignment in the dermis are quantified over 7 mm size FOV, revealing distinct spatial and temporal variations of the vascular profiles during skin remodeling.

## Results

### Layer-specific visualization of the dorsal skin

The uLSOM system used an excitation beam at 532 nm and 558 nm wavelengths, which was focused onto the skin surface by a lens mounted thru a spherically focused ultrasound (US) transducer (Fig. 1a). This scan head was immersed in water for acoustic coupling and moved laterally to acquire volumetric US and OA datasets. The low 0.05 numerical aperture (NA) of the focusing lens enables volumetric imaging with long depth of field (DOF) extending over more than 1 mm. The transducer operated in pulse-echo mode during US scan, and subsequently in receive-only mode during OA scan. We used SKH1 mice in this study because of the lack of hair and skin pigmentation as well as an intact immune system[15]. Five mm diameter full-thickness excisional wounds, comprising all skin layers and the underlying *panniculus carnosus*, were generated in the back skin (Fig. 1b) and monitored at day 4, 7, and 10 after wounding (Fig. 1c).

To record baseline parameters and verify imaging performance of the uLSOM system, we first visualized healthy dorsal skin of the same group of mice 3 days prior to wounding (day −3, Fig. 1c). As shown in Fig. 2a, pulse-echo US imaging delineates the curved skin surface by detecting reflected US waves at the interface between the skin and coupling medium, while OA imaging visualizes the multi-scale microvascular network by detecting tiny vibrations induced by optical absorption of hemoglobin. The accurate delineation of the curved skin surface, together with the long DOF, facilitate large-scale imaging of the dorsal skin, whose surface can only be flattened over a small FOV. The complementary volumetric datasets acquired with these two modalities are registered in depth, with different colors indicating the relative distance to the transducer. Based on the delineated skin surface and by further assuming a thickness of 200 μm of the dermal layer[19,20], microvasculature in the dermis and hypodermis can be segmented. Figure 2b illustrates the skin layer segmentation featuring an expanded view from Fig. 2a, where a small caliber dermis vessel (red dot) and a hypodermis vessel with bigger size (yellow dot) were selected. The corresponding 1D profiles (A-line signals) are shown on the right. The depth-registered 1D reflection US signal profile (blue dot location) was then overlaid onto the vascular signals. The skin surface (start of the epidermis) was detected as the first prominent reflection in the US signal, and the assumed 200 μm dermal thickness (gray area) discerned the dermis vessel from hypodermis vessel. By applying such segmentation on all the A-line signals, the microvasculature projected from all depths was visualized separately for the dermis and hypodermis (Fig. 2c). The microvascular network in the dorsal skin appears in distinct morphological features specific to each layer. In the dermis, a tortuous capillary network is clearly depicted, with individual vessels of diameters between 8 and 30 μm. In the hypodermis, blood vessels appear to be longer and with diameters going from a few tens to a few hundred micrometers. When the same microvascular network was color-coded with $sO_2$ levels, the morphologically homogenous blood vessels were clearly distinguished based on their roles in oxygen transport (Fig. 2d). In the dermis, feeding arterioles carrying almost fully oxygenated blood ($sO_2$ close to 100%) are interconnected with capillaries and venules with lower oxygenation ($sO_2$ between 30% and 75%). Different capillaries appear to have distinct $sO_2$ levels, potentially resulting from flow velocity-related oxygen extraction heterogeneities[21]. In the hypodermis, intertwined artery and vein pairs were clearly depicted, with arteries having close to 100% $sO_2$ and veins having $sO_2$ levels between 50% and 80%. The uLSOM system thus resolves physiologically meaningful $sO_2$ levels in large-scale microvascular network at capillary resolution in a skin layer-specific manner.

### Longitudinal monitoring of dorsal skin wound healing

We then monitored the microvascular oxygenation dynamics over the course of 10 days with a 3–4-day interval (Fig. 3). As shown in the photographs in Fig. 3a, wound areas gradually shrinked, and the wounds were significantly smaller at day 7 and 10 compared to day 4 ($p < 0.05$, Fig. 3b). The original wound area was calculated as 5 mm diameter circle, serving as an approximation of the fresh lesion at the day of wounding (day 0). The

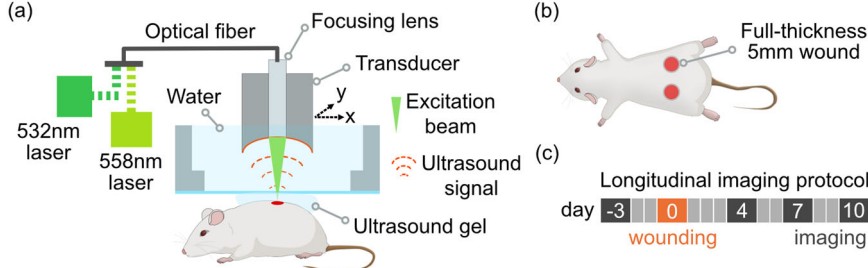

**Fig. 1 | Imaging setup, wounding and longitudinal imaging protocol. a** Simplified schematic of the imaging system. Visible laser light at 532 nm and 558 nm was used to excite OA signals from hemoglobin inside blood vessels and to further estimate hemoglobin oxygenation. Single-mode optical fiber integrated with a focusing lens was used to guide the light onto the murine skin. The focusing lens was mounted inside a spherically-focused US transducer, which forms the scan head. The transducer was used in receive-only mode for OA imaging and in pulse-echo mode for US imaging. During imaging, the scan head was immersed in water and laterally moved over the skin. US gel was applied between the water bath and skin for acoustic coupling. **b** Wounding protocol. Two full-thickness wounds with 5 mm diameter were generated after anesthesia in the shaved caudal dorsal skin with disposable biopsy punches. **c** Longitudinal imaging protocol. Baseline measurement on healthy dorsal skin was performed 3 days prior to wounding. After wounding at day 0, the same wounds were monitored at day 4, 7 and 10.

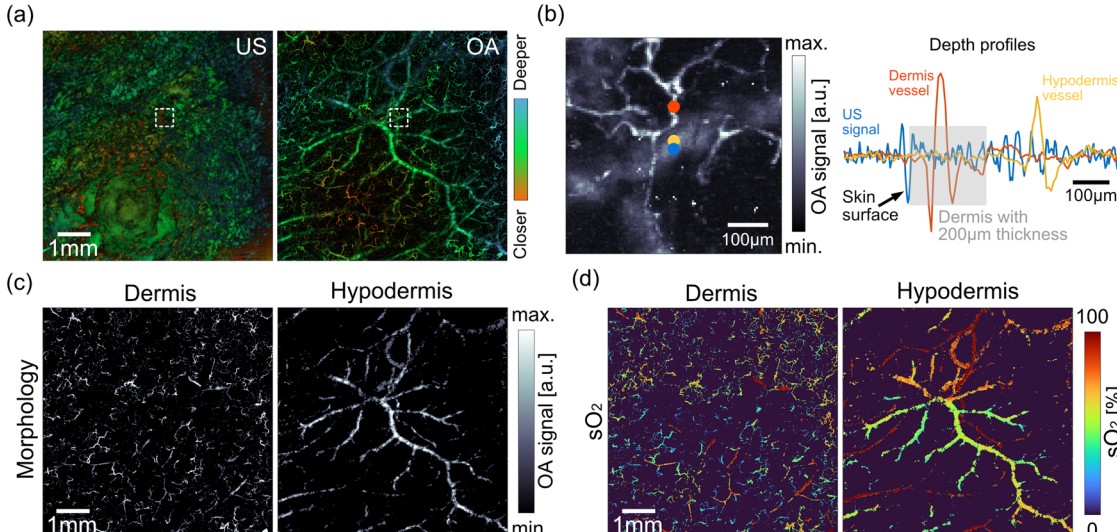

**Fig. 2 | Layer-specific visualization of microvascular morphology and sO₂. a** The imaging system acquires volumetric pulse-echo US and OA images that are registered in depth. The same colors in both modalities indicate tissue structures located at the same distance to the transducer. The curved skin surface is delineated in the US image. Microvasculature in different skin layers is visualized in OA image. Both images were taken with 7 mm × 7 mm FOV. **b** Illustration on the segmentation of blood vessels in the dermis and hypodermis. A zoom-in OA image indicated by the white dashed box in (**a**) is shown, where a small-caliber dermis vessel (red dot) and a large-diameter hypodermis vessel (yellow dot) were selected. Blue dot indicates the position of the US A-line signal. The line plot shows the overlaid US and OA A-line signals, with the assumed dermis thickness of 200 μm indicated by gray area. **c** Segmented microvascular network in the dermal and hypodermal layers. **d** The same microvasculature as in (**c**) with color-coded sO₂ levels.

wound areas were manually delineated as areas lacking vessel signals in OA images, and the boundaries were indicated by white solid lines (Fig. 3c–e). The photographs of the wound healing progression over time (Fig. 3a) corroborated the delineated wound areas. At day 4, blood vessels in the dermis manifest as a tortuous network with an elevated sO₂ level over the entire FOV (7 mm × 7 mm, Fig. 3c). As the wound closes, the vessels become more aligned towards the wound center as previously described[15], while being less oxygenated from a global perspective (day 7 and day 10, Fig. 3d, e). The same set of feeding arterioles in the dermis were consistently visualized at all time points (white arrows in Fig. 3c–e), even though capillaries and venules underwent major morphological and oxygenation changes at each time point. In the hypodermis, the morphology of large artery and vein pairs remain mostly unaltered, with new blood vessels gradually occupying the newly formed granulation tissue. Arterial sO₂ levels remain close to 100% at all time points, while the changes in venous sO₂ levels closely resemble the changes in sO₂ levels of capillaries and venules in the dermis, indicating the oxygen extraction dynamics in perfused tissues. Zoom-in images (2.5 mm × 2.5 mm) close to the wound boundary present a detailed view on the changes in morphology and oxygenation of capillaries and venules in the dermis (Fig. 3f). White arrows point to the same arteriole visualized over time, and its relative location in the images shifts to the left as the wound closes. Similar to the trend observed in global images, capillaries and venules around the indicated arteriole remodel at each time point, with more tortuous appearance at day 4, and they gradually align towards the wound center at day 7 and day 10. Concurrent with the structural changes, the capillaries and venules carry less oxygen over time, indicating a more efficient oxygen extraction of the surrounding tissue as the wound heals.

## Quantitative evaluation of spatiotemporal microvascular dynamics

As the angiogenesis during wound healing is driven by capillary sprouting[5], we focused on the quantitative analysis of the microvascular network in the dermis and granulation tissue. In addition to the longitudinal imaging capability, our technique further enables the quantification of vascular parameters as a function of the distance to wound boundary. This spatial analysis scheme is illustrated in Fig. 4a with a maximum intensity projection

(MIP) image of the dermal vessels at day 4. The full FOV (7 mm × 7 mm) was divided into five different bands. Specifically, bands 1 to 4 are concentric to the wound boundary, each has a width of 0.5 mm. Band 5 covers the remaining pixels outside band 4. This spatial division scheme was selected by considering the trade-off between maintaining enough bands to quantify spatial variations and a sufficient number of vessels within each band (e.g., tens to a few hundred) to accurately represent each distance to the wound boundary. By taking the median sO₂ level within each band, a heatmap representing the spatial gradient of microvascular sO₂ is shown in Fig. 4b. Because the wound area contains little vessel signal, pixel values within this area were discarded. The sO₂ levels gradually decrease as the distance to wound boundary increases, with median sO₂ levels at 74%, 66%, 63%, 63% and 60% in bands 1 to 5, respectively. This spatially decreasing trend was statistically confirmed (box plot in Fig. 4c), with a significant drop in sO₂ levels ($p < 0.05$) from band 1 (within 0.5 mm from the wound boundary) to peripheral bands (beyond 0.5 mm from the wound boundary). The higher sO₂ levels in band 1 at day 4 may be an indicator of the increased blood supply close to an early wound in response to the increased metabolic demand during new tissue formation[3]. The spatial variations in sO₂ levels at all time points are presented in the line plot in Fig. 4c. A trend towards a spatial decrease was consistently observed at all time points, with the most prominent changes happening at day 4 and the sO₂ levels returning to the baseline (43%, represents average sO₂ level in healthy skin, see "Methods") at day 10. Oxygenation changes are accompanied by structural adaptations. We also quantified vessel diameter and its irregularity, tortuosity and angular alignment in the same way. At day 4, vessel diameters in band 1 are significantly larger than bands 3 to 5 ($p < 0.05$, box plot in Fig. 4d), potentially accommodating the enhanced blood perfusion, especially close to the wound boundary. The line plot in Fig. 4d presents an overview of the spatial changes in vessel diameter at all time points. Much like for the oxygenation changes, a consistent spatial decrease was observed. The diameters remain larger than the baseline level (11.6 μm) at day 4 and day 7, and gradually decrease to the baseline at day 10. Vessel diameter irregularity was quantified as the standard deviation of diameter values within each band. As shown in the line plot in Supplementary Fig. 1a, vessel sizes are most irregular close to the boundary of early wounds, which was confirmed by a

**Fig. 3 | Visual tracking of microvascular sO₂ dynamics during dorsal skin wound healing.**
**a** Photographs of the imaged wound taken at day 4, 7, and 10. **b** Quantification of wound area reduction over time ($n = 6$ wounds from 4 mice). Original wound area was calculated as the area of a 5 mm diameter circle. **c–e** Longitudinal monitoring of microvascular sO₂ separately for the dermal and hypodermal layers. Wound areas were manually delineated as areas lack of OA signals and are indicated with white solid lines. White arrow heads indicate the same set of arterioles in the dermis visualized over time. The FOV for all images is 7 mm × 7 mm. **f** Zoom-in views of microvascular sO₂ in the dermis close to the wound boundary at the locations indicated with white dashed boxes in (**c**)–(**e**). White arrow heads point to the same arteriole visualized over time. The FOV for zoom-in images is 2.5 mm × 2.5 mm. Asterisks (*) indicate significant differences ($p < 0.05$).

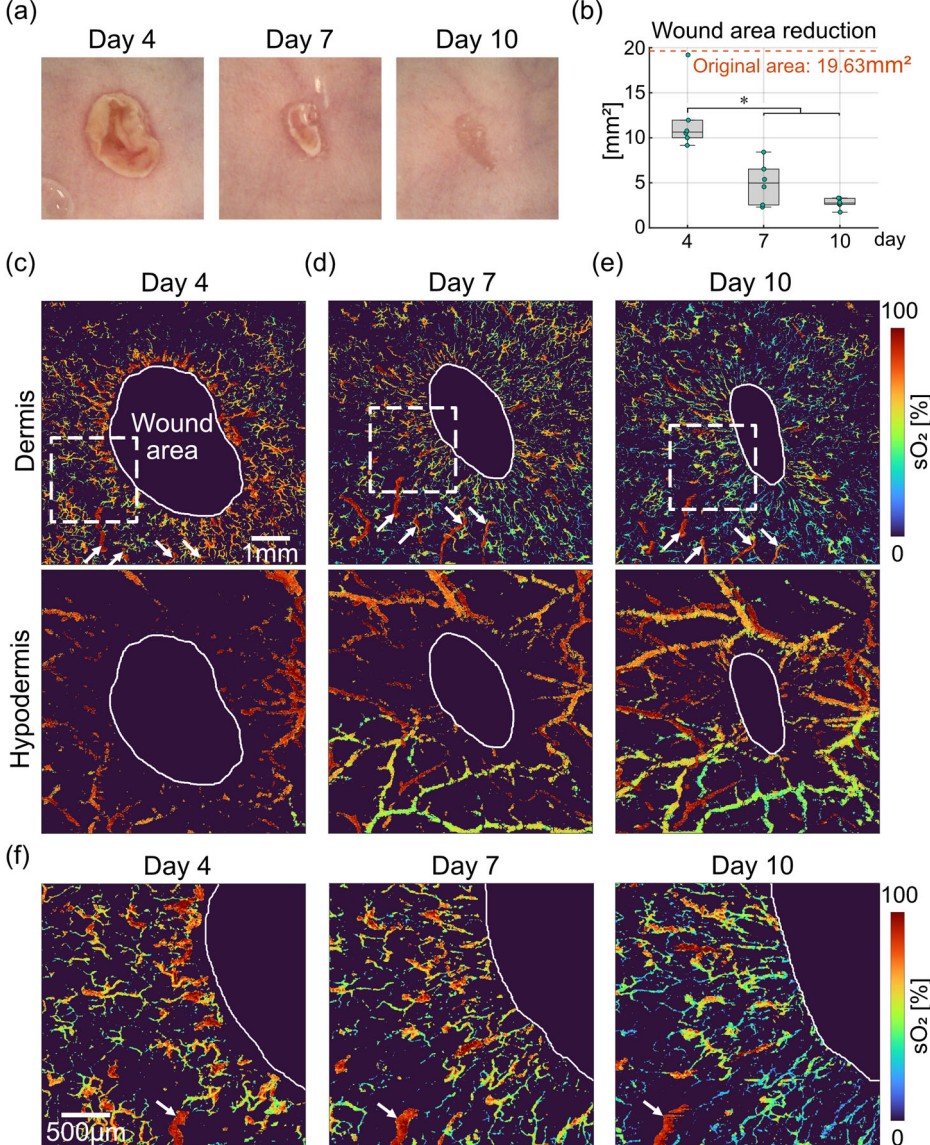

significantly higher diameter irregularity within band 1 than peripheral bands at day 4 ($p < 0.05$, box plot in Supplementary Fig. 1a). Vessel tortuosity (angular change per unit length) within band 1 is significantly higher than bands 2 to 4 at day 4 ($p < 0.05$, box plot in Fig. 4e), and an overview of its spatial changes at all time points is presented in the line plot. The spatial changes in angular alignment towards the wound center are considerably different from the trends seen in oxygenation, diameter and tortuosity (line plot in Supplementary Fig. 1b). Because the angular alignment was calculated with respect to the wound center, it is not applicable for baseline measurement on the healthy skin. A clear spatially decreasing trend was observed from bands 3 to 5 (beyond 1.5 mm from the wound boundary), and this decrease is significant at day 10 ($p < 0.05$, box plot in Supplementary Fig. 1b). The line plot also shows that the vessels become more aligned towards the wound center over time, consistently in all bands.

Besides the spatial variations, temporal changes of the same set of vascular parameters were also quantified. At the healing front (i.e., band 1, within the first 0.5 mm to the wound boundary, Fig. 5a), sO₂ levels at day 10 are significantly lower compared to day 4 and day 7 ($p < 0.05$, first box plot in Fig. 5b). The sO₂ levels remain higher than the baseline level of 43% at day 4 and day 7, and fall back to the baseline at day 10. The decreasing sO₂ levels over time may be a result of stabilizing blood perfusion and more efficient oxygen extraction as the wound heals. The temporal changes in vessel

diameter show a noticeable decrease (middle box plot in Fig. 5b), while the changes between different time points were not significant. The irregularity of vessel diameters shows a significant decrease at day 7 and day 10 compared to day 4 ($p < 0.05$, right box plot in Supplementary Fig. 1a), suggesting that the vessels around early wounds are characterized by irregular sizes[22]. A mild decrease in vessel tortuosity can be observed over time (right box plot in Fig. 5b), while no significant changes were detected between different time points. The temporal changes were further quantified in the entire granulation tissue, which is the new tissue that fills the wound[23]. Here, we approximated the fresh wound at day 0 as a 5 mm diameter circle placed at the geometric center of the wound boundary at each time point (Fig. 5c). The granulation tissue is then defined as the area that fills the fresh wound, which enlarges over time. Much like for the healing front, sO₂ levels in the entire granulation tissue show a temporal decrease, with significantly lower sO₂ levels at day 7 and day 10 compared to day 4 ($p < 0.05$, first box plot in Fig. 5d). Significant decreases in vessel diameter were also detected between each time point ($p < 0.05$, middle box plot in Fig. 5d). The blood vessels in the entire granulation tissue are thus characterized by significantly higher sO₂ levels and larger diameters at day 4, and they gradually carry less oxygen while normalize in size over time. Similar to the healing front, vessel tortuosity within the granulation tissue did not show significant changes over time (right box plot in Fig. 5d).

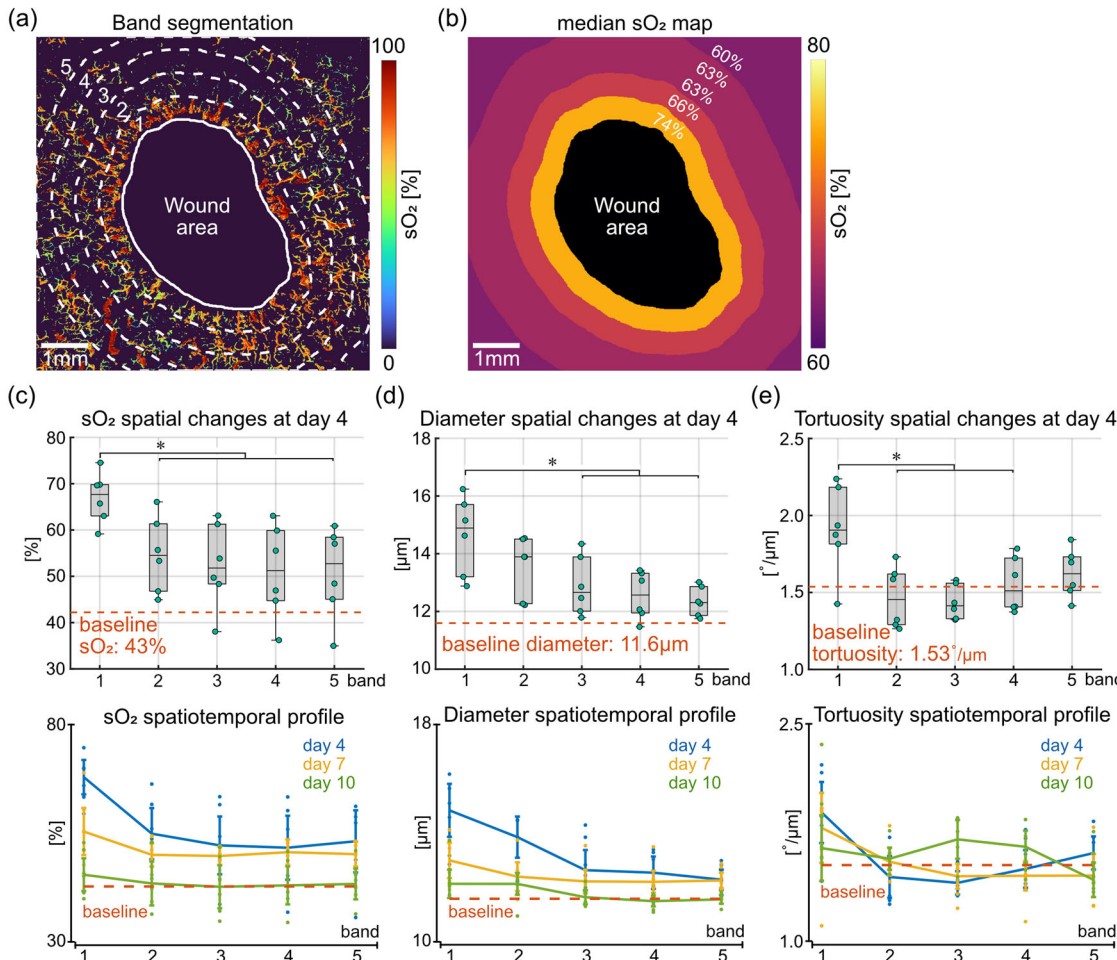

**Fig. 4 | Quantitative analysis of the spatial variations in microvascular parameters in the dermis. a** Segmentation of different spatial bands away from the wound boundary. Bands 1 to 4 are concentric to the wound boundary and have a width of 0.5 mm. Band 5 covers the remaining pixels outside band 4. The full FOV is 7 mm × 7 mm. **b** Heatmap with colors indicating median $sO_2$ within each band. Pixel values within the wound area were discarded. **c** Spatial profile of microvascular $sO_2$ changes. Box plot shows the spatial changes at day 4 ($n = 6$ wounds from 4 mice). Line plot presents an overview of the spatial changes at all time points. Baseline measurement was taken on healthy skin of the same group of mice. **d** Spatial profile of vessel diameter changes. **e** Spatial profile of vessel tortuosity changes. Asterisks (*) indicate significant differences ($p < 0.05$). Error bars in line graphs are defined as 95% confidence intervals.

## Discussion

We present a longitudinal study to assess the dynamics of microvascular oxygenation and structural adaptions during wound angiogenesis in the dorsal mouse skin. Direct assessment of wound angiogenesis has been achieved by uniquely combining high resolution depth-resolved oxygenation sensing with centimeter-scale FOV in a label-free and non-invasive fashion, thus overcoming key technical challenges of other skin imaging approaches. This imaging capacity distinguishes our work from intravital fluorescence microscopy studies that can only cover a limited FOV further necessitating exogenous contrast agents[8,24–26]. Optical coherence microangiography is capable of large-scale and label-free imaging, but lacks the oxygenation information within perfused blood vessels[12]. The uLSOM system enables layer-specific visualization of functional cutaneous microvasculature and offers comprehensive quantification of the spatiotemporal profiles of wound angiogenesis. We found that blood vessels in the dermis are more oxygenated at the healing front, and the oxygenation gradually decreases as moving away from the wound boundary. This trend is particularly prominent at an early stage of wound healing (e.g., day 4). Interestingly, this observation is consistent with a recent study on calvaria bone defect healing[27], where significantly higher $sO_2$ levels were also found around the leading edge of the calvarial defect. Longitudinally, a trend towards a decrease in microvascular $sO_2$ was observed from day 4 through day 7 until day 10, covering the new tissue formation phase of the wound

healing process[5]. This trend was consistently observed across a large spatial scale (7 mm × 7 mm), and is consistent with data from a previous OA microscopy study reporting elevated $sO_2$ levels around early wounds and its decline over time[18]. We also observed that vessel diameter changes follow a similar spatiotemporal trend as the oxygenation changes, with significantly larger diameters observed in early wounds (at day 4). The enlarged vessels gradually mature and normalize in size over time[22]. Furthermore, our quantitative analysis shows that vessels around early wounds have larger size variations and are more tortuous than vessels further away from the wounds, resonating well with the previous observation of a ring of circular vessels next to the wound boundary exhibiting large and irregular diameters[22]. In contrast to the trends seen in oxygenation and diameter, vessel angular alignment consistently increased over time, indicating a more directed blood supply towards the wound center. Collectively, our observations indicate that wound angiogenesis initiates as active capillary sprouting with enlarged newborn vessels and elevated oxygenation around the wound edge, and the angiogenic vessels gradually normalize in size and oxygenation during remodeling, to be more aligned towards when the wound is healed.

Previous OA microscopy studies on wound healing in mouse models mainly targeted the mouse ear[18,28], which is more accessible from an imaging point of view, but less clinically relevant[15]. Indeed, imaging the curved and highly scattering dorsal skin robustly over time presents a

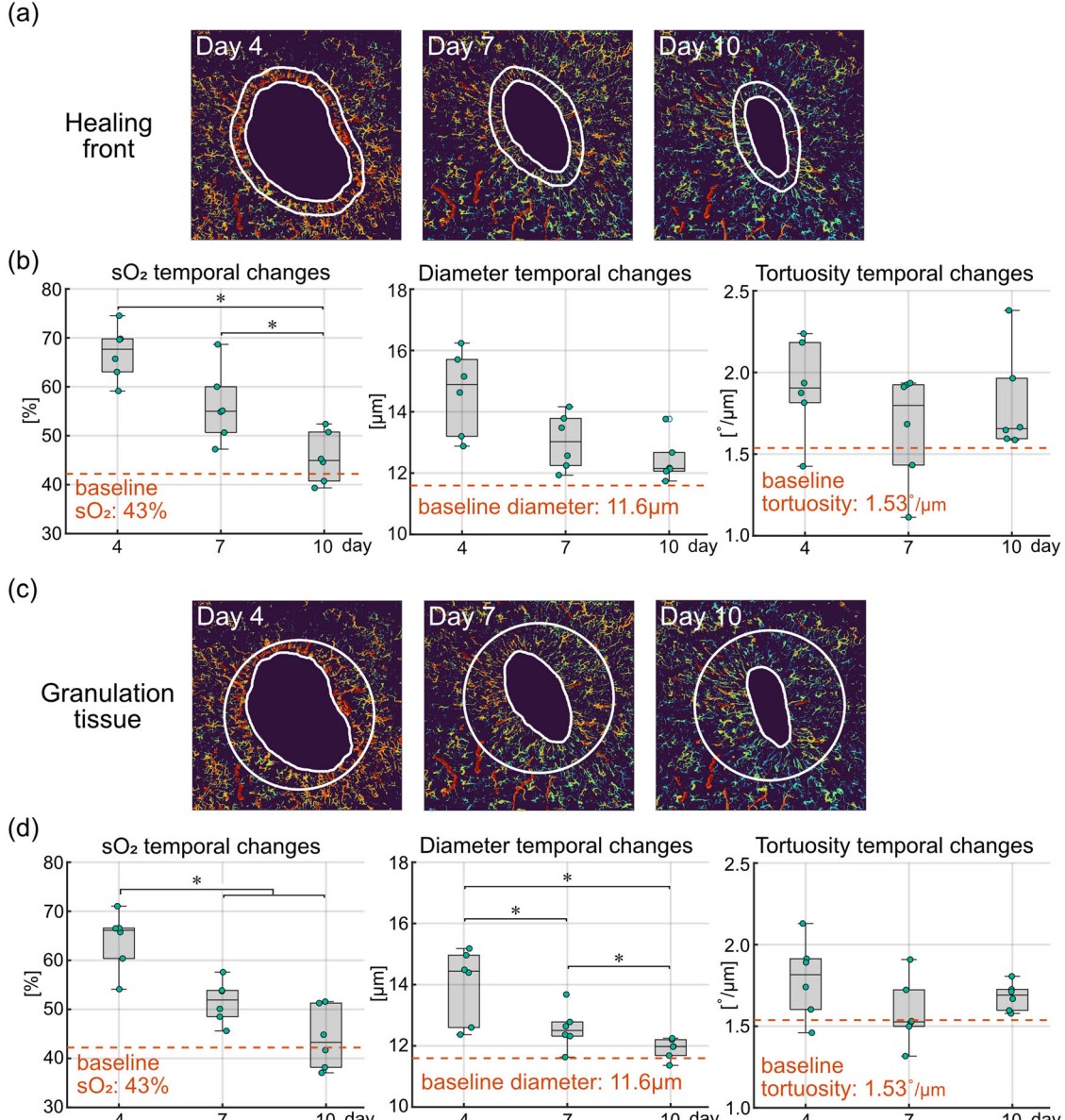

**Fig. 5 | Quantitative analysis of the temporal variations in microvascular parameters at the healing front and in the entire granulation tissue. a** Illustration of the healing front, i.e., the first 0.5 mm from the wound boundary. **b** Quantification of temporal changes in microvascular sO₂, vessel diameter and tortuosity at the healing front ($n = 6$ wounds from 4 mice). Baseline measurement was taken on healthy skin of the same group of mice. **c** Illustration of the granulation tissue, defined as the new tissue that fills the wound. The fresh wound at day 0 was approximated as the white circle with 5 mm diameter located at the geometric center of the wound boundary at each time point. Granulation tissue is the area filling the fresh wound, which enlarges over time. **d** Quantification of temporal changes in microvascular sO₂, vessel diameter and tortuosity in the entire granulation tissue ($n = 6$ wounds from 4 mice). Baseline measurement was taken on healthy skin of the same group of mice. Asterisks (*) indicate significant differences ($p < 0.05$).

technical challenge, which was largely overcome by developing a rapid large-scale OA microscopy system in combination with a flexible dorsal imaging mount[15]. In contrast to previous works[15,16], our study is the first to report on depth-resolved label-free oxygenation imaging and comprehensive assessment of spatiotemporal oxygenation changes in dorsal wounds. For this, we incorporated spectroscopic (dual-wavelength) measurements and developed a dedicated processing workflow to compensate for wavelength-dependent light fluence variations, thus providing physiological sO₂ information that is a key indicator of oxygen delivery during wound healing. We further utilized pulse-echo US information to segment microvasculature from different skin layers and analyzed wound angiogenesis in a layer-specific manner. While other OA microscopy studies have assessed the wound healing process on a

relatively small (sub-millimeter) scale[18], our approach has enabled quantifying the distinct spatial and temporal profiles on a large scale covering entire 5 mm diameter wounds, thus offering direct observations on how the microvascular network adapts its structure and function to support the healing process.

Apart from the oxygenation and structural information, another important functional parameter of blood vessels is the flow velocity. By implementing a decorrelation-based signal analysis scheme[29] or combining the system with e.g., OCM capable of blood flow quantification[30], the link between blood transit time and sO₂ heterogeneity within the capillary network may be deciphered[21]. Concurrent measurement of blood flow and sO₂ may further enable the estimation of metabolic rate of oxygen in the microvasculature[31]. Furthermore, analysis of the pulse-echo US signals

would enable the quantification of complementary parameters of the investigated skin, including thickness and echogenicity[32].

The skin layer segmentation procedure assumes a constant dermal thickness of 200 μm, which represents an average value for the mouse sex and strain used in this study[19]. Due to changes in the thickness of skin layers during skin remodeling, this assumption may not always be accurate. The dermal thickness may thus be finetuned for individual mice or time points based on the depth-registered US and OA information. Note that our goal is to achieve faithful vessel segmentation, not the accurate delineation of skin layer boundaries.

The longitudinal imaging time points were selected to investigate the new tissue formation phase of wound healing (2–10 days after injury), during which new blood vessels form[5]. The starting time point (day 4) corresponds to the onset of the angiogenesis process[24], with the imaging experiments carried out at 3-day intervals according to the approved animal protocol. The investigated time frame only allowed us to determine vascular changes until the onset of the tissue remodeling phase, whilst additional time points may be included in future studies to evaluate the later stages[5].

In summary, we expect the presented study to boost the applicability of the uLSOM system in the broad field of skin research, facilitating longitudinal monitoring of the vasculature in various skin diseases and evaluating therapeutic efficacy in unperturbed living tissues.

## Methods

### Imaging system
The uLSOM system is designed for high-resolution large-scale imaging of the microvasculature (Fig. 1a). To generate OA signals primarily from optical absorption of hemoglobin, laser light in the visible spectrum was used. Specifically, 532 nm and 558 nm light was employed to enable blood vessel structure and oxygenation imaging based on high hemoglobin absorption and distinct spectra of oxygenated and deoxygenated hemoglobin at these two wavelengths[33]. The 532 nm wavelength was provided by a pulsed nanosecond laser (Onda ns Q-switched DPSS laser at 532 nm, Bright Solutions, Prado, Italy). The 558 nm wavelength was provided by a dye laser (Credo, Sirah Lasertechnik, Grevenbroich, Germany) pumped with a nanosecond pulsed laser operating at 532 nm (Model: IS80-2-L, EdgeWave, Wuerselen, Germany). Both light paths were guided by a single-mode optical fiber with a gradient index (GRIN) focusing lens (GRIN-TECH, Jena, Germany) attached at the other end. The GRIN lens was mounted in a thru hole of a single-element spherically focused US transducer (Precision Acoustics, Dorchester, UK) with ~25 MHz central frequency and ~100% effective bandwidth, and these two components form the scan head. The GRIN lens focused the laser beams to a diffraction-limited spot with a diameter of 7 μm, which defines the lateral resolution of the uLSOM system. The optical and acoustic focus was confocally aligned to achieve optimal detection sensitivity. The transducer operated in receive-only mode to detect OA signals and in pulse-echo mode to acquire US signals, which was controlled by a pulser receiver (Model: DPR500, JSR Ultrasonics, Imaginant, USA). Before imaging, the scan head was immersed in warm water, and the water bath was gently brought in contact with the US gel for acoustic coupling. During imaging, the scan head was laterally moved over the dorsal skin to acquire a time-resolved OA/US signal at each scan position, thus recording a volumetric OA/US image. Specifically, the scan head moved rapidly along the X axis in a sinusoidal pattern with a voice-coil stage (X-DMQ12P-DE52, Zaber Technologies, Vancouver, Canada), and continuously moved along the Y axis with a linear stage (LNR50SEK1/M, Thorlabs). The US and OA scans were performed sequentially without moving the mouse, therefore the images from both modalities are spatially registered.

### Skin layer segmentation
The depth-registered OA and US datasets enabled volumetric segmentation of different skin layers, which can be generally classified as the epidermis, dermis and hypodermis[19]. As we focused on microvascular analysis and the epidermis is thin and known to be non-vascularized[5], the skin was segmented into the dermis and hypodermis in this study. By taking advantage of the large acoustic impedance mismatch between the skin surface and acoustic coupling medium (US gel), the skin surface can be delineated by identifying the first prominent reflection layer in US images. Next, the dermal layer was segmented by assuming a thickness of 200 μm[19,20]. Therefore, by moving the skin surface 200 μm down, the dermis-hypodermis junction can be delineated. Since focused excitation light in the visible spectrum was used, an effective penetration depth of around 500 μm can be achieved in the dorsal murine skin. The hypodermal layer was then segmented by taking the remaining detectable depth range in OA images, i.e., around 300 μm.

### Vessel parameter estimation
Blood $sO_2$ can be estimated from spectroscopic OA images by exploring the spectrally distinctive optical absorption of oxygenated and deoxygenated hemoglobin and performing spectral unmixing based on the measured OA spectrum[34]. In practice, the spatially and spectrally varying light fluence before and after light-tissue interaction corrupt the measured OA spectrum, which needs to be corrected to reveal physiologically meaningful $sO_2$ levels[34]. We partitioned fluence variations into incident fluence variations (i.e., before light-skin interaction) and local fluence variations (i.e., after light-skin interaction). Incident fluence variations between the two excitation wavelengths were monitored with a high-speed photodiode (DET36A2, Thorlabs) on a pulse-by-pulse basis and corrected by scaling the corresponding OA signals[35]. The wavelength-dependent local fluence variations were further partitioned into variations caused by skin reflectance spectrum and those caused by attenuation along depth. The reflectance spectrum of the murine dorsal skin has previously been characterized[19,20] to monotonically decrease within a narrow 525–575 nm spectral window. We estimated that light at 532 nm is reflected around 1.1 times more than at 558 nm, and applied this scaling factor to the A-line signals to correct for wavelength-dependent skin reflectance. Fluence attenuation through different skin layers can be modeled as an exponentially decaying function by selecting with wavelength-specific absorption and scattering coefficients[36]. In our implementation, we have only modeled the fluence attenuation in the extravascular space before the light reaches a vessel. The attenuation length is then defined by the skin surface and the vessel depth, i.e., the axial location corresponding to the maximum OA signal amplitude. We then set the effective attenuation coefficient to be the reduced scattering coefficient of the skin, assuming that optical attenuation in the extravascular space is dominated by scattering[36]. Note that this assumption may not be valid for the epidermis due to high optical absorption by melanosomes, but it does not introduce significant modeling errors because the epidermis is very thin (less than 20 μm) while the SKH1 mice used have little skin pigmentation. By performing spectral unmixing on fluence-corrected OA spectrum, physiologically meaningful blood $sO_2$ levels can be quantified.

Apart from the oxygenation level, structural parameters were also estimated from the OA images, including vessel diameter and its irregularity, tortuosity and angular alignment. Vessel diameter can be estimated based on a previously reported automatic vessel segmentation and analysis algorithm[15], which provides precise morphological features of the vessel segments such as start and end points, center line and edges. Diameter irregularity was calculated as the standard deviation of individual diameter values within a specific region. The morphological features of vessel segments further enabled the estimation of tortuosity and angular alignment. Vessel tortuosity is defined as the angular change of a vessel segment per unit length (e.g., 1 μm). Vessel angular alignment is defined as the angle deviation of a vessel segment with respect to the wound center[15]. The angular alignment values were normalized to fall in the range between −1 and 1, with 1 representing full alignment and −1 representing perpendicular orientation. Baseline values of all vascular parameters were obtained on the healthy skin of the same group of mice used for wound monitoring by calculating the median of each pre-wound FOV and the mean of all pre-wound FOVs.

## Animal handling, wounding and longitudinal imaging protocol

Female SKH1 hairless mice (9–10 weeks old; Charles River Laboratories, Germany) were housed in a ventilated cage in a temperature-controlled room under a 12-h dark-light cycle (temperature 21 ± 1°C, relative humidity 55 ± 10%). Pelleted food and water were provided ad libitum. Mice were anaesthetized using ketamine/xylazine injection prior to wounding. Two full-thickness excisional wounds, including the epidermis, dermis, sub-cutaneous adipose tissue and the underlying *panniculus carnosus* with diameters of 5 mm were generated in the caudal dorsal skin with disposable biopsy punches (Fig. 1b). Three days prior to wounding (day −3), imaging of the healthy dorsal skin was performed under isoflurane anesthesia, serving as the baseline measurement (Fig. 1c). The wounds were then left to heal without dressing. They were longitudinally monitored at 4, 7 and 10 days post injury under isoflurane anesthesia. Imaging was not performed at day 0 due to mild bleeding. All imaging sessions followed the same protocol. Firstly, a pulse-echo US scan was performed over 7 mm × 7 mm FOV at a step size of 20 µm and completed in 45 s. Then, an OA scan was performed over the same FOV at a step size of 5 µm and completed in 9 min. During the scan, the mouse head was kept still using a stereotactic head holder (SG-4N; Narishige, Tokyo, Japan), and the dorsal skin was gently compressed with a custom-made dorsal imaging mount[15] to eliminate breathing-induced motion artifacts. Throughout the experiment, none of the animals showed signs of disturbances in eating and drinking habits, and all wounds healed without complications. Animal experiments had been performed in accordance with the Swiss Federal Act on Animal Protection and approved by the Cantonal Veterinary Office Zurich. We have complied with all relevant ethical regulations for animal use.

## Statistical analysis

Quantification of microvascular parameters was performed on MIP images of the dermal layer. Median and interquartile range are indicated in the box plots ($n = 6$ wounds from 4 mice), with median shown as black line inside the boxes and interquartile range shown as the length of the boxes. For longitudinal data, the paired *t* test was used to test for significance after the sample distribution passed the Shapiro–Wilk normality test. When comparing different bands at a specific time point (unpaired data), the non-parametric Kruskal–Wallis test was performed first to detect if any significant difference exists. If a significant difference was detected, the Wilcoxon rank sum test was subsequently performed to assess the difference between pairs of samples. All *p* values obtained from the above significance tests were further adjusted for multiple testing to correct for the false discovery rate using the Benjamini and Hochberg method[37,38]. Differences were considered significant when a *p* value was lower than 0.05.

## Reporting summary

Further information on research design is available in the Nature Portfolio Reporting Summary linked to this article.

## Data availability

Source data for the charts and graphs in the main and Supplementary Figs. is available as Supplementary Data. The raw datasets and images are available for research purposes from the corresponding author upon request.

## Code availability

The specific implementation of the data processing and analysis code, together with example datasets for testing the shared code, are available on the Zenodo repository website at: https://doi.org/10.5281/zenodo.14062345[39].

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

## Acknowledgements
D.R. and S.W. are members of the collaborative research consortium SKINTEGRITY.CH. We thank E. Jessen from Technical University of Darmstadt for suggestions in band segmentation.

## Author contributions
W.L. and Y.H.L. performed imaging experiments. F.K. performed wounding experiments. W.L. performed data analysis. S.W. and D.R. supervised the study. All authors contributed to preparing figures, writing and editing the manuscript.

## Funding

## Competing interests
The authors declare no competing interests.
