## [Transparent Peer Review file · Communications Biology]

Skin Layer-Specific Spatiotemporal Assessment of Micrometabolism during Wound Angiogenesis

Corresponding Author: Mr Weiye Li

This manuscript has been previously reviewed at another journal that is not operating a transparent peer review scheme. The manuscript was considered suitable for publication without further review at Communications Biology.

Version 0:

Reviewer comments:

Reviewer #1

(Remarks to the Author)

In the article, the authors described a hybrid imaging methodology with pulse-echo ultrasounds and spectroscopic optoacoustic microscopy. They assessed the changes in vascularization and oxygenation in mice dermis and hypodermis during wound healing.

The subject is very relevant for the field. The work here is original. The state-of-art and the problematic are clearly presented in the introduction. The results are neatly displayed, and the data analysis is coherent. However, some pieces of information could be added to gain clarity in the experimental design, and to go a step further in the discussion.

1. Please precise in the abstract that this is an in vivo study, using mice as a model.
2. Similarly, add information about your biological model at the end of your introduction (line 61-62 for example).
3. It is unclear for me how your technology helped you answer to your claims: compared to the literature a bigger field of view, an increase in depth oxygenation imaging, without exogenous agent.
 - a. Please provide precisions about the utility of the pulse-echo ultrasound. Is it only to delineate the start of the epidermis?
 - b. Please elaborate about how you reached a bigger field of view?
4. The biopsy punch removes the first layers of the skin. How deep are the fresh wounds? After a few days of healing, the wounds close, but the skin layer are rebuilding, and the thickness of the skin layers may be impacted.
 - a. How did you delineate the start of the epidermis?
 - b. Could you elaborate on your assumption of the 200 μm limit in this case?
5. How many mice were used in total? Is your $n=6$ on 3 mice? Are the controls from prior wounding or from independent mice?
6. 3 time-points are limited to conclude on time-dependent tendencies. Please elaborate on this choice and its implications in the discussion.
7. I find the words "long-term study" slightly misleading. 10 days is biologically not very long.
8. The discussion could benefit from some rewriting to underline comparisons between your work and the literature, outside of its originality that was already presented in introduction. For example, the comparison with bone defect healing (reference 25) is very interesting. Could you precise somewhat the resonance of your work with references 26, and other references on wound healing neo-vascularization, alignment and oxygenation?

Reviewer #2

(Remarks to the Author)

The author studied full-thickness excisional wounds at different healing stages with hybrid ultrasound and spectroscopic optoacoustic microscopy. Skin layer-specific vascularization is visualized at capillary resolution over centimeter-scale field-of-view in a non-invasive, label-free manner. Different vascular parameters, including oxygenation, diameter and angular alignment, show distinct spatial and temporal variations. The article is sufficiently clear in most points. The following few comments are therefore aimed at strengthening and clarifying the point.

1. The author needs to explain the laser setting parameters and pulse echo setting parameters.
2. Why divide it into five different bands? On what basis was this decision made?
3. Considering the design of Fig 4, should adding the curve from days 0 -3 to make a more comprehensive comparison with on day 4?
4. The author can use a colorbar with better contrast to more clearly display the difference between high oxygen saturation and low oxygen saturation.
5. The article (line 111-112) states that "The same set of feeding arterioles in the dermis were consistently visualized in the lower left corner of the images at all time points, in contrast to the fact that capillary morphology and oxygenation changed vastly at each time point." The author should specify which feeding arterioles are specified in the Fig 3.
6. The author should add some statistical analysis, for example, describing the reduction in wound area and the tortuosity of capillaries in Fig 3.
7. Why does the author still think there is a difference (line 160-161) even though there is no statistical difference between the two?

Reviewer #3

(Remarks to the Author)

The authors proposed a study of full-thickness excisional wound healing using dual-modality ultrasound and optoacoustic microscopy. Different vascular parameters, such as oxygenation, diameter, and angular alignment were investigated on day 4, day 7, and day 10 after wounding. However, the major concern about this manuscript is its innovation. The optoacoustic microscopy has been extensively developed over the last two decades. There are various published studies of wound healing using optoacoustic microscopy. The OAM system used in this paper has been published before and was also used in a similar study of wound healing in the authors' previously published work (reference 16). The author claimed that imaging over centimeter-scale FOV is challenging, but in this paper, they didn't specify what innovation has been made in their technology or method to achieve a larger FOV over other optoacoustic microscopy techniques. Many published works provide large FOV optoacoustic microscopy techniques that can achieved over centimeter-scale FOV imaging (e.g. Chen, J., Zhang, Y., He, L., Liang, Y., & Wang, L. (2020). Wide-field polygon-scanning photoacoustic microscopy of oxygen saturation at 1-MHz A-line rate. *Photoacoustics*, 20, 100195.; Qin, W., Jin, T., Guo, H., & Xi, L. (2018). Large-field-of-view optical resolution photoacoustic microscopy. *Optics express*, 26(4), 4271-4278.; Chen, M., Jiang, L., Cook, C., Zeng, Y., Vu, T., Chen, R., ... & Yao, J. (2022). High-speed wide-field photoacoustic microscopy using a cylindrically focused transparent high-frequency ultrasound transducer. *Photoacoustics*, 28, 100417.; Baik, J. W., Kim, J. Y., Cho, S., Choi, S., Kim, J., & Kim, C. (2019). Super wide-field photoacoustic microscopy of animals and humans in vivo. *IEEE transactions on medical imaging*, 39(4), 975-984.). It will be helpful if the authors can show what type of significant scientific question they can address with this work.

Minor:

1. What is the temporal resolution of this OAM imaging? Is the temporal resolution high enough to obtain oxygen saturation in living animals?
2. A major advancement the authors claimed in this work is that they extended the imaging capacity of the previous system to incorporate oxygenation information. What did the author do to extend the ability of the previous system?
3. How did the authors resolve the problem of animal breathing during the scan? The animals may move as they breathe, does it affect the imaging?
4. There should be some more details of the scanning, for example, what does it use to scan the probe? Does it use linear stage, MEMS, or other methods? What is the step size and how long does it take to acquire an image? Is there any new scanning mechanism to achieve a larger FOV? We need some information on how it achieves a larger FOV than other OAM techniques since this is a major advancement the authors claimed in this paper.
5. What is the scientific significance of the findings in this work? What important new knowledge it can provide us in addition to the previous research on wound healing?

Version 1:

Reviewer comments:

Reviewer #1

(Remarks to the Author)

My questions have been answered. I have no further comment.

Reviewer #2

(Remarks to the Author)

I thank the authors for their thoughtful and comprehensive answers to each of my points. I read with great interest.

The author is very clear about the colorbar settings. The difference of the image may also be due to the contrast of the original image affecting the blood oxygen results after calculation.

Reviewer #3

(Remarks to the Author)

The author has answered my questions and made corresponding changes. I have no further questions. The revised article meets the requirements of the journal, and I recommend it for publication.

Reviewer 1

In the article, the authors described a hybrid imaging methodology with pulse-echo ultrasounds and spectroscopic optoacoustic microscopy. They assessed the changes in vascularization and oxygenation in mice dermis and hypodermis during wound healing.

The subject is very relevant for the field. The work here is original. The state-of-art and the problematic are clearly presented in the introduction. The results are neatly displayed, and the data analysis is coherent. However, some pieces of information could be added to gain clarity in the experimental design, and to go a step further in the discussion.

Reply: We thank the reviewer for the positive evaluation of our work and providing valuable suggestions to help improve the manuscript. Please find below a point-by-point response to each comment.

1. Please precise in the abstract that this is an in vivo study, using mice as a model.

Reply: We have specified in the abstract that this is an in vivo study performed on mouse dorsal skin (page 1, lines 18-19).

2. Similarly, add information about your biological model at the end of your introduction (line 61-62 for example).

Reply: We have clarified at the end of the Introduction that the wounds were generated on the dorsal skin of mice (page 2, line 63).

3. It is unclear for me how your technology helped you answer to your claims: compared to the literature a bigger field of view, an increase in depth oxygenation imaging, without exogenous agent.

a. Please provide precisions about the utility of the pulse-echo ultrasound. Is it only to delineate the start of the epidermis?

b. Please elaborate about how you reached a bigger field of view?

Reply: We thank the reviewer for the suggestion to improve the clarity of our methodology. At the beginning of the Discussion section, we meant to emphasize that our technology offers a unique combination of large field of view (FOV), depth-resolved oxygenation sensing and label-free imaging. It is the combined imaging features that set our work apart from previous literature, not a particular feature alone, e.g. large FOV. We have edited the corresponding description to improve clarity (page 7 lines 215-222).

The pulse-echo ultrasound (US) is indeed used to delineate the start of the epidermis. We have prepared an illustration on the use of pulse-echo US and skin layer segmentation, and it is included as panel (b) of the revised Figure 2 (see below). The skin surface (start of the epidermis) is detected in the US signal as the first prominent reflection. Please refer to our reply to comment 4 below for a detailed explanation on the use of pulse-echo US for skin layer segmentation.

Regarding the FOV, we would like to clarify that the bigger FOV is in comparison with previous confocal/two-photon microscopy studies. There are other OA microscopy systems that can also provide large FOV. In our system, the centimeter-scale FOV is achieved via

scanning the focusing lens and US transducer together with rapid mechanical stages. This design is in contrast to systems employing a stationary objective whose FOV is inversely related to the numerical aperture (NA). For high-NA systems, big FOV can only be achieved by moving the imaged sample and performing lateral stitching, which is typically time-consuming. Furthermore, specific to the dorsal skin imaging scenario, the FOV can additionally be limited by the requirement of a flat skin surface to achieve optimal image quality, and the skin can only be made very flat over a small FOV. In contrast, our system uses pulse-echo US to delineate the curved skin surface over large FOV, and the long depth-of-field (enabled by low NA = 0.05) ensures good image quality over depth variations of more than 1mm. We have elaborated on the system design facilitating large imaging FOV in the revised manuscript (page 3 lines 75-77, 88-90).

4. The biopsy punch removes the first layers of the skin. How deep are the fresh wounds? After a few days of healing, the wounds close, but the skin layer are rebuilding, and the thickness of the skin layers may be impacted.

a. How did you delineate the start of the epidermis?

b. Could you elaborate on your assumption of the 200 µm limit in this case?

Reply: We thank the reviewer for raising the above questions. The wounds at day 0 comprise the complete epidermis, dermis, subcutaneous adipose tissue and underlying panniculus carnosus (for thickness see reference [1] below). We have added this information in the revised manuscript (page 3 lines 80-81, page 11 lines 378-379).

As mentioned in the response to comment 3, an illustration on the use of pulse-echo US to delineate skin surface and facilitate skin layer segmentation is included in panel b of the revised Figure 2 (see figure above). The start of the epidermis was delineated as the first prominent reflection in the US signal. However, the dermis-hypodermis junction is difficult to delineate from the trailing peaks. Based on the depth-registered US and optoacoustic (OA) datasets, the OA signals originating from vessels in the dermis and hypodermis can be overlaid onto the US signal. Then, by assuming a dermal thickness of 200µm [1,2] (gray area), the blood vessels in the dermis and hypodermis can be segmented. We agree with the reviewer that the thickness of the skin layers is changing while they are rebuilding, implying thickness heterogeneities over

space and also over time. However, our goal is to achieve faithful vessel segmentation, not the accurate delineation of the boundary of each skin layer. From our experience, the assumed constant dermal thickness of 200 μ m worked robustly over the whole FOV (as demonstrated in Figure 2c) and also across the different mice used in this study. To obtain a more accurate estimation for each individual mouse, the assumed dermal thickness may be further finetuned and validated as demonstrated in Figure 2b. We have elaborated on the discussion of the assumption of dermal thickness in the revised manuscript (page 8 lines 271-276).

[1] C.P. Sabino et al., "The optical properties of mouse skin in the visible and near infrared spectral regions", *Journal of Photochemistry and Photobiology B: Biology*, Volume 160, July 2016, Pages 72-78.

[2] K. Calabro et al., "Gender variations in the optical properties of skin in murine animal models", *Journal of Biomedical Optics*, Vol. 16, Issue 1, 011008 (January 2011).

5. How many mice were used in total? Is your n=6 on 3 mice? Are the controls from prior wounding or from independent mice?

Reply: A total of 4 mice were used. We generated 2 wounds on each mouse, giving a total of 8 wounds. We eventually used the data from 6 out of 8 wounds because the other 2 wounds were challenging to position under the system properly at one time point, making vessel quantification prone to experimental error. The 2 discarded wounds were from 2 different mice. Therefore, we have n=6 wounds from 4 mice. We have added this information in the Methods section (page 11 line 396) and also in corresponding figure legends. The controls/baseline measurements are from the same group of mice prior to wounding, i.e. at day -3. We have clarified the control/baseline measurement in the revised manuscript (page 10 lines 370-372).

6. 3 time-points are limited to conclude on time-dependent tendencies. Please elaborate on this choice and its implications in the discussion.

Reply: We thank the reviewer for this suggestion. Days 4, 7, 10 post wounding were chosen to focus our analysis on the new tissue formation phase (2-10 days after injury) of wound healing, in which new blood vessels form [1]. The 3-day time interval was set according to our experimental license - the mice are only allowed to be anesthetized every 3 days to comply with animal welfare requirements. Therefore, we started the longitudinal imaging at day 4 (onset of the angiogenesis process [2]) and continued at days 7 and 10. We acknowledge that the selected time points are limited to conclude on the tendencies of vascular changes beyond the new tissue formation phase, and more time points after day 10 may be included in future studies to evaluate the remodeling phase of wound healing [1]. We have commented on the choice of the time points in the revised manuscript (page 8 lines 277-283).

[1] G. C. Gurtner, S. Werner, Y. Barrandon, & M. T. Logaker, Wound repair and regeneration, *Nature*, 453, 314-321 (2008).

[2] M. K. Schneider, H-I Ioanas, J. Xandry & M. Rudin, *Scientific Reports*, 9, 6004 (2019).

7. I find the words "long-term study" slightly misleading. 10 days is biologically not very long.

Reply: We thank the reviewer for pointing out this misleading description. By "long-term" we meant to emphasize that our imaging study was performed longitudinally on the same group of mice *in vivo*. We have changed the word "long-term" to "longitudinal" throughout the revised manuscript.

8. The discussion could benefit from some rewriting to underline comparisons between your work and the literature, outside of its originality that was already presented in introduction. For example, the comparison with bone defect healing (reference 25) is very interesting. Could you precise somewhat the resonance of your work with references 26, and other references on wound healing neo-vascularization, alignment and oxygenation?

Reply: We thank the reviewer for the suggestion to enrich the discussion. We have elaborated on the comparison between our observations with findings in the literature. Please refer to the revised manuscript (page 7 lines 232-240) for the detailed discussion.

Reviewer 2

The author studied full-thickness excisional wounds at different healing stages with hybrid ultrasound and spectroscopic optoacoustic microscopy. Skin layer-specific vascularization is visualized at capillary resolution over centimeter-scale field-of-view in a non-invasive, label-free manner. Different vascular parameters, including oxygenation, diameter and angular alignment, show distinct spatial and temporal variations. The article is sufficiently clear in most points. The following few comments are therefore aimed at strengthening and clarifying the point.

Reply: We thank the reviewer for the positive evaluation of our manuscript and the constructive suggestions on how to improve it. Please find below the point-by-point response to each comment.

1. The author needs to explain the laser setting parameters and pulse echo setting parameters.

Reply: We thank the reviewer for pointing out the insufficiently precise description of the imaging parameters. We have specified the device models for the lasers, transducer and pulser-receiver, and also the scan settings for pulse-echo US and spectroscopic OAM in the revised manuscript (page 9 lines 295-299, 302-303, 308-309; page 11, lines 384-387).

2. Why divide it into five different bands? On what basis was this decision made?

Reply: We thank the reviewer for raising this question. We decided to define a width of 0.5mm for each band by considering a trade-off between the number of bands within the field-of-view (FOV) and the number of vessels covered within each band. The number of bands should be sufficiently high to allow the quantification of spatial variations, and at the same time, each band should cover a sufficient number of vessels to accurately represent each distance to the wound edge. Given the imaging FOV of 7mm x 7mm and the average wound size of 11mm² at day 4, defining a band width of 0.5mm yields 5 different bands, with each band containing tens to a few hundred vessels. The band width of 0.5mm also represents an approximation of the healing front with respect to the wound size (in our case it is 10% of the original wound diameter). We have elaborated on the choice of this band division scheme in the revised manuscript (page 4 lines 152-155).

3. Considering the design of Fig 4, should adding the curve from days 0 -3 to make a more comprehensive comparison with on day 4?

Reply: We thank the reviewer for this suggestion. We acknowledge that adding data at day 0 will make the comparison with the following time points more comprehensive. However, we did not perform imaging at the day of wounding (day 0) because the fresh wounds tend to bleed, which may significantly deteriorate the image quality and introduce quantification errors. The data from day -3 was included as the indicated baseline measurement (orange dashed lines). Since day -3 only involves imaging of the healthy skin, the segmentation of different bands with respect to the wound boundary is not applicable. Therefore, we included day -3 by taking the median of the each individual pre-wound FOV, and further taking the mean of all pre-

wound FOVs (n=6). Thus, the baseline represents the average vascular parameter of the healthy skin of all pre-wound areas. We have commented on the complications of imaging at day 0 (page 11 lines 383-384) and elaborated on the description of baseline measurement at day -3 in the revised manuscript (page 10, lines 370-372). We have also updated the design of the original Figure 4 by separating spatial and temporal quantifications in the revised Figure 4 and 5 respectively.

4. The author can use a colorbar with better contrast to more clearly display the difference between high oxygen saturation and low oxygen saturation.

Reply: We thank the reviewer for this suggestion. In terms of color scheme selection, we actually tried a number of different colormaps and found the current one (turbo colormap in Matlab) to give the best contrast between high and low oxygen saturation while being visually pleasant (e.g. compared to jet colormap). Regarding the color limit of the colorbar, we tried to set the lower limit to e.g. 40%, but this leads to the loss of some capillary/venule signals in the images. Therefore, the colorbar settings remain unchanged. We would be happy to learn about further suggestions on the colorbar settings from the reviewer.

5. The article (line 111-112) states that "The same set of feeding arterioles in the dermis were consistently visualized in the lower left corner of the images at all time points, in contrast to the fact that capillary morphology and oxygenation changed vastly at each time point." The author should specify which feeding arterioles are specified in the Fig 3.

Reply: We thank the reviewer for pointing out this unclear reference to Fig. 3. We have included arrows in panel (c) of the revised Fig. 3 to indicate the same set of feeding arterioles present at all time points. We have also adapted the corresponding description in the revised manuscript (page 4 lines 128-130).

6. The author should add some statistical analysis, for example, describing the reduction in wound area and the tortuosity of capillaries in Fig 3.

Reply: We thank the reviewer for this suggestion. We have added statistical analysis on the reduction in wound area in panel (b) of the revised Fig. 3. Due to space limit, we included the statistical analysis on vessel tortuosity in the revised Fig. 4 and Fig. 5. The original angular alignment analysis was moved to panel (b) of the new supplementary figure S1. The corresponding description was also updated in the revised manuscript (page 4 lines 118-124).

7. Why does the author still think there is a difference (line 160-161) even though there is no statistical difference between the two?

Reply: We thank the reviewer for pointing out this description that needs to be improved. We have reformulated the corresponding description (page 5 lines 195-196), and also expanded the description on vessel diameter changes based on the revised Fig. 5.

Reviewer 3

The authors proposed a study of full-thickness excisional wound healing using dual-modality ultrasound and optoacoustic microscopy. Different vascular parameters, such as oxygenation, diameter, and angular alignment were investigated on day 4, day 7, and day 10 after wounding. However, the major concern about this manuscript is its innovation. The optoacoustic microscopy has been extensively developed over the last two decades. There are various published studies of wound healing using optoacoustic microscopy. The OAM system used in this paper has been published before and was also used in a similar study of wound healing in the authors' previously published work (reference 16). The author claimed that imaging over centimeter-scale FOV is challenging, but in this paper, they didn't specify what innovation has been made in their technology or method to achieve a larger FOV over other optoacoustic microscopy techniques. Many published works provide large FOV optoacoustic microscopy techniques that can be achieved over centimeter-scale FOV imaging (e.g. Chen, J., Zhang, Y., He, L., Liang, Y., & Wang, L. (2020). Wide-field polygon-scanning photoacoustic microscopy of oxygen saturation at 1-MHz A-line rate. *Photoacoustics*, 20, 100195.; Qin, W., Jin, T., Guo, H., & Xi, L. (2018). Large-field-of-view optical resolution photoacoustic microscopy. *Optics express*, 26(4), 4271-4278.; Chen, M., Jiang, L., Cook, C., Zeng, Y., Vu, T., Chen, R., ... & Yao, J. (2022). High-speed wide-field photoacoustic microscopy using a cylindrically focused transparent high-frequency ultrasound transducer. *Photoacoustics*, 28, 100417.; Baik, J. W., Kim, J. Y., Cho, S., Choi, S., Kim, J., & Kim, C. (2019). Super wide-field photoacoustic microscopy of animals and humans in vivo. *IEEE transactions on medical imaging*, 39(4), 975-984.). It will be helpful if the authors can show what type of significant scientific question they can address with this work.

Reply: We thank the reviewer for pointing out the above references, and apologize for the confusion. We would like to clarify that we did not mean to claim larger FOV over previous optoacoustic microscopy (OAM) techniques, and agree with the reviewer that there are numerous reported OAM techniques that offer an imaging FOV bigger than the presented study. In fact, we meant to emphasize that it is challenging to achieve centimeter-scale FOV, depth-resolved oxygenation sensing, and capillary-resolution imaging at the same time, not imaging centimeter-scale FOV alone. Actually, it is the unique combination of the above mentioned imaging features that sets our work apart from previous intravital microscopy studies employing not only OAM, but also fluorescence microscopy and optical coherence microscopy. We have clarified this issue in the revised manuscript (page 7 lines 215-222).

Our previous work (reference 16, [1] below) focused on evaluating the effects of overexpression of VEGF-A and comparing with wild-type control mice, whereas the presented study focuses on spatiotemporal analysis of the angiogenesis process, especially the oxygenation aspect, on the same group of mice. The OAM system employed in [1] uses a single wavelength at 532nm to image the vascular structure, which is not capable of estimating blood oxygenation based on spectroscopic measurement. Therefore, the key distinguishing factor of the presented study is the comprehensive assessment of spatiotemporal microvascular oxygenation dynamics during wound healing. Also, reference [1] lacks the pulse-echo ultrasound information that is critical to delineate the curved skin surface, segment skin layers and enable layer-specific analysis, as demonstrated in the presented study.

We also thank the reviewer for raising the question of what type of significant scientific question can the presented study address. To the best of our knowledge, our work is the first to comprehensively assess the spatiotemporal changes in microvascular oxygenation during wound healing in the dorsal skin. The distinct spatial and temporal profiles were quantified at a larger scale than ever reported, offering direct observations on how the microvascular network adapts its structure and function to support the healing process. The current study demonstrates the feasibility of monitoring the entire large-diameter (5mm) wounds in the dorsal skin, which constitutes a more biologically relevant target than small-diameter (0.8mm) wounds in the mouse ear [2]. Thus, it sets the stage for future studies answering questions related to oxygen delivery dynamics in pathological skin conditions and during pharmacological interventions.

We have commented on the difference between the current study and our previous work and elaborated on the scientific question that our study can address in the revised manuscript (pages 7-8, lines 251-262).

[1] Y-H Liu et al., Non-invasive longitudinal imaging of VEGF-induced microvascular alterations in skin wounds, *Theranostics* 12, 558-573 (2022).

[2] N. Sun et al., Photoacoustic microscopy of vascular adaptation and tissue oxygen metabolism during cutaneous wound healing, *Biomedical Optics Express*, Vol. 13, Issue 5, pp. 2695-2706 (2022).

Minor:

1. What is the temporal resolution of this OAM imaging? Is the temporal resolution high enough to obtain oxygen saturation in living animals?

Reply: We thank the reviewer for this question. For an imaging FOV of 7mm x 7mm with 5 μ m step size, the OAM scan took 9 minutes. We acknowledge that our technique offers a temporal snapshot of the microvascular oxygen saturation (sO_2), not real-time monitoring of sO_2 changes. However, since we are investigating the temporal changes on the scale of days, the 9-minute temporal resolution is sufficient to obtain the sO_2 information. We have included the scan duration in the revised manuscript (page 11 lines 385-387).

2. A major advancement the authors claimed in this work is that they extended the imaging capacity of the previous system to incorporate oxygenation information. What did the author do to extend the ability of the previous system?

Reply: We thank the reviewer for this question. The imaging capacity was extended from single-wavelength imaging of vascular structure in our previous study [1] to dual-wavelength imaging of vascular structure and oxygenation in the current study. Apart from incorporating a second laser source and associated optical components into the system hardware, we further developed an sO_2 estimation workflow to account for wavelength- and depth-dependent fluence variations caused by light-skin interaction. The upgraded system thus provides depth-resolved, physiologically meaningful sO_2 maps. We have extended the description on our methodology in the revised manuscript (page 10, lines 341-356).

[1] J. Rebling et al., Long-term imaging of wound angiogenesis with large-scale optoacoustic microscopy, *Advanced Science* 2021, 8, 2004226.

3. How did the authors resolve the problem of animal breathing during the scan? The animals may move as they breathe, does it affect the imaging?

Reply: We thank the reviewer for raising this important question. Firstly, a stereotactic head holder was used to hold the mouse head still throughout the scan. Furthermore, a 3D printed dorsal imaging mount [1] was used to gently press the dorsal skin area against the heating pad, making the imaging area free of breathing motion artifacts. We have provided detailed information on the head holder and dorsal imaging mount in the revised manuscript (page 11 lines 387-390).

4. There should be some more details of the scanning, for example, what does it use to scan the probe? Does it use linear stage, MEMS, or other methods? What is the step size and how long does it take to acquire an image? Is there any new scanning mechanism to achieve a larger FOV? We need some information on how it achieves a larger FOV than other OAM techniques since this is a major advancement the authors claimed in this paper.

Reply: We thank the reviewer for the suggestions to clarify our methodology. Firstly, we did not mean to claim larger FOV over other OAM techniques, and please refer to our reply to the major comment for a detailed clarification on our claims. We have provided detailed information on the scanning mechanism and parameters in the revised manuscript (page 9 lines 312-315; page 11, lines 385-387).

5. What is the scientific significance of the findings in this work? What important new knowledge it can provide us in addition to the previous research on wound healing?

Reply: We thank again the reviewer for this question. Please refer to our reply to the major comment for a detailed clarification. We hope the reviewer finds that the clarity of the revised manuscript significantly improved to be considered for publication in *Communications Biology*.